# Evidence of *Xylella fastidiosa* Infection and Associated Thermal Signatures in Southern Highbush Blueberry (*Vaccinium corymbosum* Interspecific Hybrids)

**DOI:** 10.3390/plants12203562

**Published:** 2023-10-13

**Authors:** Melinda Guzman Martinez, Jonathan E. Oliver, Paul M. Severns

**Affiliations:** 1Department of Plant Pathology, University of Georgia, Athens, GA 30602, USA; 2Department of Plant Pathology, University of Georgia, Tifton, GA 31793, USA

**Keywords:** *Xylella fastidiosa*, thermal imaging, bacterial leaf scorch, pre-symptomatic disease detection

## Abstract

*Xylella fastidiosa*, a gram-negative bacterium vectored to plants via feeding of infected insects, causes a number of notorious plant diseases throughout the world, such as Pierce’s disease (grapes), olive quick decline syndrome, and coffee leaf scorch. Detection of *Xf* in infected plants can be challenging because the early foliar disease symptoms are subtle and may be attributed to multiple minor physiological stresses and/or borderline nutrient deficiencies. Furthermore, *Xf* may reside within an infected plant for one or more growing seasons before traditional visible diagnostic disease symptoms emerge. Any method that can identify infection during the latent period or pre-diagnostic disease progress state could substantially improve the outcome of disease control interventions. Because *Xf* locally and gradually impairs water movement through infected plant stems and leaves over time, infected plants may not be able to effectively dissipate heat through transpiration-assisted cooling, and this heat signature may be an important pre-diagnostic disease trait. Here, we report on the association between thermal imaging, the early stages of *Xf* infection, and disease development in blueberry plants, and discuss the benefits and limitations of using thermal imaging to detect bacterial leaf scorch of blueberries.

## 1. Introduction

Plant diseases are nearly ubiquitous throughout the world, and their diversity appears to be substantially underestimated outside of agricultural systems [1,2]. In general, plant diseases become noticeable when they either directly or indirectly generate conspicuous problems, such as the demise of a dominant or keystone species in natural ecosystems [3,4,5] or an indirect perturbation to animal populations directly linked to plants impacted by disease [6,7]. However, it is the impact of plant diseases on agricultural systems that receives the majority of plant pathologists’ research attention and focus.

Early interventions into plant and animal disease outbreaks can either suppress or prevent an epidemic (a form of biological invasion) from developing [8,9,10,11]. Disease detection, especially early in an outbreak (or any biological invasion), is essential for precise delineation of the infected population and effective disease outbreak suppression. Delayed biocide interventions require larger footprints to offset treatment delays, exposing pest populations to more broadly distributed purifying selection, which has generated fungicide [12,13], pesticide [14], herbicide [15], and/or antibiotic resistance in pest organisms [16,17]. Early detection and treatment appear to be key in the control of disease outbreaks because the area required to eradicate or even suppress an outbreak increases exponentially with increasing time from the epidemic onset [11]. It is possible that smaller-scale, early outbreak treatments may delay the emergence of biocide-resistant diseases through a decreased treatment footprint, and repeated applications may be unnecessary if the infected population is treated early enough for an outbreak to be suppressed [11].

*Xylella fastidiosa* (*Xf*, a name inclusive of all named subspecies), a gram-negative bacterium, is an important and potentially devastating plant pathogen. Like other plant pathogens that cause systemic disease, the earliest visible symptoms of *Xf* infections can be very subtle (slight leaf discoloration, <1 cm diameter brown regions, rolled, and browning leaf margins). These same symptoms may be readily confused with abiotic stresses, such as water deficit, fertilizer and/or chemical burns, insect herbivory, and mechanical damage [18]. *Xf*-caused diseases are managed like many other systemic plant diseases. The disease management strategy is to remove the obviously infected plants and to consider removing plants within a buffer zone that could be latently infected [19,20]. Because *Xf*-caused diseases are typically slow to develop (at least one growing season), vectoring of *Xf* to uninfected plants via feeding by flying, hemipteran insects (primarily Cicadellidae and Aphrophoridae [21,22,23,24,25]) almost assuredly spreads *Xf* to uninfected hosts far beyond the immediate disease outbreak area. Unfortunately, the spatial patterns of disease transmission are difficult to understand and, therefore, manage due to a combination of poorly understood insect vector behaviors and a long disease latent period. *Xf* causes disease in hundreds of different plant species throughout the warmer, temperate regions of the world [26], including culturally important plants, such as grapes (Pierce’s disease), olives (olive quick decline syndrome), and coffee (coffee leaf scorch) [27,28]. So, there is a general interest in methods to detect disease as early as possible, which could improve plant removal strategies to more effectively limit disease spread. In the southeastern United States, bacterial leaf scorch (BLS) of blueberry is caused by *Xf* subsp. *multiplex* and *Xf* subsp. *fastidiosa* [29,30]. BLS causes significant yield loss and plant death in this region, which produces a large proportion of the spring North American fresh market blueberries. There are no curative treatments for BLS (e.g., antibiotics), and BLS appears typical among *Xf*-caused diseases in having a long, potentially multiyear latent period [31,32,33].

*Xylella fastidiosa* is known to interfere with the translocation of water in infected host plant tissues [34,35], but disease symptoms are typically first expressed in leaves. For BLS, the disease can be visibly diagnosed by symptoms including a combination of marginal leaf scorching, significant leaf drop, and yellowed stems [36]. The leaf scorching symptom is believed to be due to insufficient water translocation throughout leaves, which have locally accumulated a sufficient density of bacteria in the vascular system. Similar to Pierce’s disease of grapes, the early stages of BLS display a patchy distribution of discolored and marginally scorched leaves intermingled with leaves appearing to be unaffected and completely healthy on the same and adjacent stems [36]. As BLS progresses, disease symptoms become more uniformly and continuously distributed within one or several stems. This eventually leads to regions of the blueberry plant displaying yellow and dead stems, which expand throughout the plant over time, ultimately resulting in whole plant death [36]. It is possible that before BLS leaf scorching symptoms appear, the plant may experience local water translocation dysfunction for days, weeks, or months. If local water movement is hindered by *Xf* prior to visible symptom expression, infected leaves may not be able to effectively dissipate heat through evapotranspiration (as a normally functioning leaf would be able to). There may be thermal (infrared) foliar symptoms that are displayed prior to the presentation of diagnostic BLS symptoms that could be useful for early disease detection.

To determine whether the early stages of BLS development produce thermal signals that may be useful for early disease detection prior to the presentation of visible diagnostic symptoms in greenhouse and a range of outdoor environmental conditions, we experimentally inoculated two different cultivars of tissue-cultured blueberry plants with *Xf* subspecies *multiplex* (*Xfm AR3* isolate ‘AlmaReb3’—hereafter *Xfm AR3 AR3*) [30] and tracked disease development for 3 months using a thermal imaging camera (Fotric 228GRD thermal imager) and traditional visual observations. If *Xfm AR3* infection interferes with leaf cooling through evapotranspiration, we expect to observe higher leaf temperatures on experimentally inoculated blueberry plants when compared to control plants, either before or coinciding with subtle foliar disease symptom appearance (slight leaf discoloration). 

## 2. Results

### 2.1. Traditional and Thermal Tracking of Disease Symptoms and Progression

Symptoms consistent with BLS were observed on inoculated plants beginning approximately 39 days after inoculation, but no such symptoms were observed on any of the mock-inoculated (control plants) (Figure 1). At 41 days post-inoculation (DPI), the mean disease severity rating (DSR) of the inoculated ‘Rebel’ plants was ~0.3 on the 0–7 DSR scale, and the inoculated ‘Emerald’ plants did not show any visible disease symptoms until 43 DPI (Figure 1). The first thermal signatures of infection (individual leaves noticeably hotter than adjacent leaves on the same stem) were observed in the inoculated ‘Emerald’ at 45 DPI and 39 DPI for ‘Rebel’. Overall, disease progression was more rapid in ‘Rebel’ than in ‘Emerald’ plants, with both subtle visual and thermal symptoms occurring in ‘Rebel’ two and six days before subtle disease symptoms were observed in ‘Emerald’ plants.

### 2.2. Leaf Temperature Readings with Disease Severity Progression

The mean thermal temperatures of *Xfm AR3* inoculated ‘Rebel’ plants were higher at all three time periods of 14, 34, and 54 days post-inoculation (Table 1) than the SCP buffer-only control ‘Rebel’ plants. Despite the variation in the minimum temperatures recorded between the ‘Emerald’ and ‘Rebel’ plants, all of the greatest recorded maximum temperatures were associated with the *Xfm AR3* inoculated plants (Table 1). The greatest recorded temperature of 37.3 °C was at 14 DPI in the *Xfm AR3* inoculated ‘Emerald’ plants, while the lowest temperature was recorded within ‘Rebel’ at 54 DPI in the SCP buffer-only controls (Table 1).

The thermal images of uninoculated plants revealed a relatively uniform distribution of heat within the mock-inoculated stems (Figure 2 and Figure 3), while leaves on the stems that were *Xfm AR3* inoculated contained leaves that were noticeably warmer than the adjacent leaves (Figure 2 and Figure 3). This variation is reflected by both the observed maximum and minimum temperatures within those inoculated stems, as well as typically smaller standard deviations for most dates where measurements were summarized (Table 1). Despite the asymptomatic leaves on the mock- and *Xfm AR3*-inoculated stems in Figure 2 and Figure 3, the *Xfm AR3*-inoculated stems displayed higher mean thermal temperatures in both cultivars at each assessment time point. Within the mock SCP buffer-only control plants, leaf temperatures were observed to be much more uniform (Figure 2 and Figure 3).

### 2.3. Thermal Disease Detection at Different Ambient Temperatures

In direct sunlight at an ambient temperature of 6 °C, the mean temperatures observed in the treatment and control plants of both ‘Emerald’ and ‘Rebel’ showed no statistically significant differences within cultivars (Figure 4). At an ambient temperature of 12 °C, *t*-tests indicated a statistically significant difference in leaf temperature between ‘Rebel’ buffer-only control and ‘Rebel’ *Xfm AR3* inoculated plants, having a mean temperature difference of 4 °C, and a significant difference was observed again when ambient temperatures were 16 °C (*t* = −5.16, *p* = 8.11 × 10^−6^) (Figure 4). Within the ‘Emerald’ plants, no significant difference was observed between the buffer-only control and *Xfm AR3* inoculated ‘Emerald’ plants at 6 °C (*t* = −0.69208, *p* = 0.49) or at 12 °C (*t* = −1.42, *p* = 0.16), but a significant difference was observed at 16 °C (*t*= −7.24, *p* = 1.14 × 10^−8^) (Figure 4).

### 2.4. Whole Plant Temperature Distribution in Control and Xfm AR3 Inoculated Plants

The arithmetic means of the *Xfm AR3* inoculated ‘Emerald’ and ‘Rebel’ (45.2 °C and 46.2 °C) plants were consistently higher than their corresponding buffer-only control plants, 42.0 °C and 42.9 °C, respectively (Table 2). Additionally, the median temperatures followed the same pattern with the *Xfm AR3* inoculated ‘Emerald’ and ‘Rebel’ plants having higher temperatures (37.6 °C and 39.0 °C) compared to their buffer-only controls (34.6 °C and 37.1 °C), respectively. Maximum leaf temperatures for the buffer-only control plants were 3.2 °C and 3.3 °C cooler in the ‘Emerald’ and ‘Rebel’ plants, respectively, than the maximum leaf temperatures of their *Xfm AR3* inoculated counterparts (Table 2). In the buffer-only control plants, the leaf temperatures appeared to be relatively uniform throughout the plant, while the *Xfm AR3* inoculated plants presented visibly hotter leaves scattered throughout different branches of the plant (Figure 5).

The minimum leaf temperature recorded was in the *Xfm AR3* inoculated ‘Emerald’ at 31.5 °C, and the maximum leaf temperature observed was in the *Xfm AR3* inoculated ‘Rebel’ at 46.2 °C (Table 2). Overall, ‘Rebel’ plants reached a higher minimum temperature than the ‘Emerald’ plants. The treatment that had the widest range of temperature was the *Xfm AR3* inoculated ‘Emerald’ plants, which had a difference of 13.4 °C between the lowest and highest recorded leaf temperatures.

### 2.5. Polymerase Chain Reaction (PCR)

Out of the 13 blueberry plants (10 ‘Rebel’ and 3 ‘Emerald’) with extracted DNA, seven (5 ‘Rebel’ and 2 Emerald) had been inoculated with *Xfm AR3*. Among these, three of the plant DNA samples successfully amplified the 733 bp PCR product of the *Xfm AR3* fragment. PCR results were negative for the 733 bp fragment in all control plants.

## 3. Discussion

Our exploratory experiments focused on examining the capabilities of the Fotric 228GRD thermal imager and its use for potentially detecting differences between *Xylella fastidiosa* subspecies *multiplex* (*Xfm AR3*) inoculated southern highbush (SHB) blueberry and uninoculated plants. This study had two main limitations, the first being the low number of SHB blueberry cultivar replicates and the second being consistent PCR amplification of the *Xfm AR3* target fragment from plant DNA extractions. The low number of replicates for each treatment was based on the availability of tissue-cultureed blueberry plants. While we would have preferred to have a fully balanced design with greater numbers of replicates in all treatments, we found patterns in thermal signatures and the visible development of BLS that occurred only on inoculated plants (not the buffer only control plants). These visual disease symptom progression observations were consistent with previous BLS studies. First, *Xfm AR3* inoculated plants displayed higher mean, median, and maximum foliar temperatures than the SCP buffer-only control plants (Figure 4). This outcome was expected because *Xf* is known to interfere with water movement throughout host plants [37], where the leaves will diminish the plant’s ability to cool convectively via open stomata in daylight and sufficient soil moisture. At the advanced stages of BLS disease, this disruption in water flow results in the leaf scorching symptom where the leaf margins are rolled, desiccated, and turned brown [38].

However, somewhat unexpectedly, when only subtle leaf discoloration (yellowing or small red spots), we recorded leaf surface temperatures that were 2 °C to 7 °C warmer than adjacent leaves on the same stem. We also observed conspicuously hot single leaves on *Xfm AR3* inoculated stems that showed no visible signs of disease (e.g., the leaves were as green as the neighboring leaves), and the adjacent leaves were markedly cooler. These conspicuously hot, singular leaves or clusters of hot multiple leaves bordered by relatively cool neighboring leaves on the same and nearby stems were not observed on any of the control plants. On whole plants that were relatively advanced in disease progression, thermal images (Figure 5) revealed scattered individual leaves that were extremely warm compared to surrounding leaves, and these hot leaves were interspersed throughout the *Xfm AR3* inoculated plants. Some of these leaves were up to 10 °C warmer than neighboring leaves, and all of these leaves regardless of the temperature indicated through thermal imagery, appeared to be green and not symptomatic on the *Xfm AR3* inoculated plants (Figure 5). We observed no isolated and scattered hot leaves on any of the control plants (Figure 5). The observed patchy distribution of conspicuously hot leaves on *Xfm AR3* inoculated plants is consistent with the patchy patterns of leaf scorching in the early stages of BLS [39] and even Pierce’s disease of grapes [40]. Although our replicate numbers were low, the patterns and expression of thermal and visual symptoms were consistent among all the inoculated plants and absent from the control plants.

Although all of our *Xfm AR3* inoculated plants developed symptoms typical of BLS, the attempts to PCR-verify the presence of *Xfm AR3* in experimental plants fell short of ideal expectations, with three of seven inoculated plants positively amplifying the 733 bp region associated with BLS. Nonetheless, visible symptoms were consistent with BLS, and conspicuously hot leaves were only observed in *Xfm AR3* inoculated plants. This inconsistent amplification of a diagnostic PCR amplicon for BLS was not entirely surprising as previous studies have demonstrated the difficulty of culturing *Xf* [41] and extracting *Xf* DNA from infected plant and insect tissues using a diversity of DNA extraction approaches [42,43,44]. Though our molecular verification of *Xfm AR3* infections for all experimental plants was not entirely successful, our use of tissue cultured nursery propagated plants and the observation that no other disease symptoms were observed other than those assignable to BLS suggests that observed visible and thermal symptoms were due to differences in plant *Xfm AR3* infection status.

Traditional methods of identifying diseased plants rely on visible symptoms, which often appear at later stages of infection. However, we present evidence suggesting that the pre-diagnostic stages of BLS from *Xfm AR3* infection may be indicated by differences in foliar leaf temperatures. This may provide a means to detect infected blueberry plants that would otherwise appear to be asymptomatic for BLS. Early detection is likely to be key in controlling BLS (and other *Xf*-caused plant diseases) because, despite *Xf* infections causing systemic plant disease [45], its distribution throughout the plant is patchy and unpredictable. The thermal imaging techniques and patterns identified in our study may allow for the application of thermal imaging technology as a non-invasive early detection method. Selectively removing diseased plants earlier in the outbreak cycle could reduce the risk of pathogen spread and suppress disease outbreaks before the disease intensifies and spreads throughout a field or across the greater landscape. Thermal imaging may also potentially benefit nurseries wishing to evaluate young blueberry plants for possible *Xf* infections. Assuming that BLS symptom development in the field is similar to our experimentally inoculated plants, thermal imaging may provide an effective means to identify diseased plants for testing or removal before they develop visible diagnostic symptoms. Our experiments with potted blueberry plants suggest that thermal diagnosis under field conditions will likely require direct sunlight and air temperatures exceeding 12 °C.

## 4. Materials and Methods

### 4.1. Experimental Design and Xylella fastidiosa Inoculation

For our experimental inoculations, plants of southern highbush (SHB) blueberry (*Vaccinium corymbosum* interspecific hybrids) cultivars ‘Emerald’ and ‘Rebel’ were acquired from Fall Creek Nursery (Fall Creek Farm Nursery Inc., Lowell, OR, USA). Cultivar ‘Emerald’ is considered to be tolerant to bacterial leaf scorch (BLS) in the field, while cultivar ‘Rebel’ develops more severe symptoms with comparatively rapid disease progression [29]. We used tissue-cultured plants to ensure that experimental plants were uninfected prior to inoculation and that any expressed disease symptoms were attributable entirely to *Xfm AR3*. The bare root tissue cultured plants were transferred into 9.5 L pots with one part sand and three parts pine bark mulch. Plants were maintained under greenhouse conditions of 28–35 °C and relative humidity of 60–90%. Plants were watered *ad libitum* and fertilized using Osmocote Smart Release Plant Food Plus fertilizer at a rate of 44 mL per 7.5 L pot [30].

*Xylella fastidiosa* subspecies *multiplex* (*Xfm AR3*) isolate ‘AlmaReb3’, a virulent *Xfm AR3* strain previously isolated and shown to consistently cause BLS in the greenhouse by Di Genova et al. [30] was used for all plant inoculations. AlmaReb3 stocks were maintained in long-term storage with 20% glycerol (at −80 °C). For experimental inoculations, AlmaReb3 was streaked onto plates of periwinkle wilt (PW) agar medium [46]. These plates were incubated at 28 °C, then subcultured after 10–14 days, and observed for isolated bacterial colony growth [47]. Pure colonies from Xfm AR3 subcultures were suspended in 1 succinate-citrate-phosphate (SCP) buffer [46]. Using a NanoDrop OneC (Thermo Scientific, Waltham, MA, USA) spectrophotometer, the concentration of the bacterial suspension and SCP buffer was adjusted until the inoculation concentration of ~1 × 10^8^ cells/mL was realized.

On 14 January 2021, we experimentally inoculated two-month-old Emerald and Rebel potted blueberry plants with ~20 µL of the bacterial *Xfm AR3* suspension into a single stem using a 1 mL tuberlin syringe with a 23-gauge hypodermic needle [30]. The single stem inoculation was performed near the base of the stem ~4 cm above the soil interface, penetrating through the phloem and into the xylem tissue as previously described by Di Genova et al. [30]. These same seven blueberry plants were re-inoculated using the previously described method seven days after the first inoculation to ensure infection. For uninoculated control plants, sterile SCP buffer was delivered using a hypodermic needle according to the method described above (n = 1 plant cultivar ‘Emerald’, n = 5 plants of cultivar ‘Rebel’). The numbers of cultivar ‘Emerald’ and ‘Rebel’ inoculated and control plants varied due to a limited number of available tissue-cultured ‘Emerald’ plants and greenhouse space.

### 4.2. Thermal Imaging Device

We used the Fotric 228GRD thermal imager (Fotric Inc., Shanghai, China) to capture thermal images of *Xfm AR3*-inoculated and uninoculated (buffer only) control blueberry plants as time elapsed from the second round of inoculations. The Fotric camera records thermal images with a 640 × 480 resolution (307,200 pixels) with a temperature sensitivity range between −20 °C and 650 °C and an accuracy of ±2 °C (outside conditions) and ±0.1 °C (within a stable room environment). The thermal imager was accompanied by a Samsung Galaxy J7 (Samsung Electronics Co., Ltd., Suwon-si, Republic of Korea) smartphone that served as a visual interface with the camera through which the proprietor supplied the imaging application LinkIR. The LinkIR application (Fotric Inc., Shanghai, China) provided a 15-color pallet to visualize thermal images where the camera could record and capture up to 1000 frames, five frames per second maximum, under a 10 h battery charge [48]. To take simultaneous normal light (human eyesight light wavelengths) and thermal images of the plants, the picture-in-picture function was used. This function produces a thermal and a natural light image of the same subject.

### 4.3. Traditional and Thermal Tracking of Disease Symptoms and Progression

Experimentally inoculated plants were monitored beginning on 3 March 2021 and tracked for visible disease development every other day for five weeks beginning two weeks after the second inoculation, then once every seven days until 11 May 2021 (for a total span of 69 days of observation but 117 days after the initial inoculations). We used a disease severity rating scale [36] (detailed in Table 3) to track the visible symptoms of BLS disease progression. On each date, we also captured thermal images of the same plants to understand how potential thermal signals and visible symptoms may relate to each other. The disease severity rating scale on the selected stems ranged from 0 to 7 (Table 3). A designation of zero indicated an inoculated stem with no visual symptoms of the disease. A rating of one indicated leaf yellowing, a rating of two depicted reddening on the leaf where it was yellow before, and a rating of three indicated when two or more leaves on the inoculated stem turned red. A rating of four indicated when leaves were partially yellow with red margins and necrotic areas developing, while a rating of five was when approximately 50% of the leaves were bright red with necrotic areas. A rating of six was given when leaves were red/brown with marginal necrosis, and lastly, a rating of seven was recorded when the entire stem and all of its leaves were severely scorched or necrotic with stem yellowing. Each inoculated stem of all *Xfm AR3* and SCP buffer-only (control) inoculated plants were rated, and their treatment means were plotted as a disease severity progression curve with days post-inoculation (DPI = days post first inoculation) on the x-axis and the mean disease severity rating (0–7) on the y-axis.

### 4.4. Leaf Temperature Readings across Disease Severity Progression

Based on the patterns of BLS disease symptom progression from the monitoring of *Xfm AR3*-inoculated plants [33] and our own observations, we selected the thermal images taken at 14, 34, and 54 days post-inoculation (DPI) as important time points to present the emergence of visible symptoms and potential thermal foliar symptoms on the inoculated stems. To generate heat stress above the ambient temperatures and standardize the thermal images within the greenhouse, we used a reflector lamp with a 125 W infrared bulb (Bongbada PAR38 120 V). The reflector lamp was clamped onto an irrigation support metal rod above the greenhouse benches, and individual plants were placed beneath the lamp approximately 35.5 cm from the reflector shade aperture for 5 min. After 5 min beneath the reflector lamp, a thermal image of the inoculated stem was captured with the Fotric camera and saved for later analysis. To evaluate whether there were potential differences in foliar leaf temperatures at 14, 34, and 54 days post-inoculation (DPI), we randomly selected 10 leaves on each of the inoculated stems from images in the traditional visual spectrum and recorded temperatures at 3 points down the mid-rib of each leaf (petiole base, mid-leaf, leaf apex) from the thermal image to calculate a mean, single-leaf temperature (the point tool function had an area of 5 × 5 pixels). Although we also had visible spectrum plant images, we first selected leaves from the thermal image, which helped blind us towards or away from leaves with subtle disease symptoms visible to the unaided eye. We calculated a mean plant temperature from all 10 leaves for each control (buffer-only) and *Xfm AR3* inoculated plant. Thermal temperatures were read with the AnalyzIR V5.0 software program. To describe the general patterns of leaf surface temperatures with increasing time since inoculation (*Xfm AR3* and buffer only) and the presentation of visible/thermal disease symptoms, we calculated summary statistics, the mean (± standard deviation), median (25th and 75th quartiles), and minimum and maximum leaf temperatures at 14, 34, and 54 DPI in the program JMP.

### 4.5. Thermal Symptom Threshold Detection over a Range of Ambient Temperatures

To approximately define a range of ambient air temperatures and environmental conditions under which we could possibly differentiate *Xfm AR3* inoculated plants (diseased) from control (non-diseased plants), we moved two ‘Emerald’ and three ‘Rebel’ *Xfm AR3*-infected plants displaying “hot leaves” and one ‘Emerald’ and three ‘Rebel’ control plants the absence of “hot leaves” into the direct sun on 7 March 2021 (52 days DPI) and captured thermal images as the ambient air temperature gradually rose from 6 °C to 16 °C. The ambient air temperature was checked using the local weather station (River Oaks-Whitehall KGAATHEN129, Athens, GA, 33.909° N, 83.364° W) and was recorded every 15 min beginning at 09:00 h until 14:00 h when the observations were terminated. The seven plants were arranged in a single row in direct sunlight, and top-view thermal images of each plant were taken every 30 min. Images were reviewed using the AnalyzIR software, and temperatures from 20 haphazardly selected leaves on each plant (based on the visible spectrum images), which included the inoculated stem and surrounding stems. We used a *t*-test (JMP) to determine whether leaves on *Xfm AR3* inoculated stems were, on average, warmer than those of control (mock-inoculated) plants at different ambient temperatures of 6 °C, 12 °C, and 16 °C under direct, unobstructed sunlight.

### 4.6. Whole Plant Temperature Distribution of Aged Control and Xfm AR3 Inoculated Plants

Once the *Xfm AR3* inoculated plants displayed traditional leaf scorching symptoms on the inoculated stem at 151 DPI (14-June-2021), a single ‘Emerald’ and ‘Rebel’ SCP buffer-only control and two ‘Emerald’ and ‘Rebel’ *Xfm AR3* inoculated plants were moved outside, into direct sunlight beginning at 10:00 h and terminating at 14:00 h (27 °C to 32 °C) to evaluate whether there were any thermal signatures at the whole plant level (as opposed to within an inoculated stem) that may be associated with subtle but potentially diagnosable BLS. Each plant was individually moved from the shade into direct sunlight for 10 min, and after 10 min, both thermal and standard images were taken. Twenty leaves from each plant were haphazardly selected (from the visible spectrum images) to record leaf surface temperature readings from the thermal images. We calculated summary statistics (JMP), including the arithmetic mean, median, minimum, and maximum temperatures, and the first and third quartiles to quantitatively represent emergent and general patterns between clearly diseased and un-diseased blueberry plants.

### 4.7. Xfm AR3 Infection Verification through Genotyping

After the greenhouse experiments were concluded, we extracted plant DNA using the Qiagen DNeasy Plant Mini kit per the manufacturer’s instructions (Qiagen Sciences Inc., Germantown, MD, USA) to confirm the presence of *Xfm AR3* in experimentally inoculated plants. We also extracted plant DNA from the SCP buffer-only control plants to ensure that these plants were *Xfm AR3*-free. Extractions were carried out according to the Qiagen kit instructions. After the extractions were performed, the extracted DNA was stored in a −20 °C freezer for later use in polymerase chain reactions (PCR). For individual PCR reactions, following the manufacturer’s instructions for 25 µL PCR reactions (New England BioLabs Inc., Ipswich, MA, USA), 2.5 µL of the DNA from the extractions was added to a 22.5 µL PCR mixture containing 15.875 µL H2O, 5 µL 5× OneTaq standard reaction buffer (NEB Inc.), 0.5 µL 10mM dNTPs (NEB Inc.), 0.5 µL 20µM RST31forward primer (5- GCGTTAATTTTCGAAGTGATTCGATTGC -3’), 0.5 µL 20 µM RST33 reverse primer (5- CACCATTCGTATCCCGGTG-3’) [37], and 0.125 µL OneTaq DNA polymerase (NEB Inc.). The PCR reaction had an initial denaturation at 95 °C for 2 min followed by 40 cycles of denaturation at 95 °C for 10 s, annealing at 60 °C for 15 s, and extending at 72 °C for 30 s, with a final 5 min extension at 72 °C before cooling down to 15 °C (hold).

Gel electrophoresis was performed in an electrophoresis chamber (Bio-Rad Laborato-ries Inc., Berkeley, CA, USA) through a 1.5% agarose gel (1× TBE buffer) stained with 2 µL of GelRed (Biotium Inc., Fremont, CA, USA) run at 90 V for 30 min with one well being loaded with 5 µL of the Quick-Load Purple 100 bp DNA Ladder (NEB Inc.). Nucleic acid bands were then visualized by exposure to blue LED light using a Gel-Bright LED Gel Illuminator (Biotium Inc., Fremont, CA, USA). The RST31/33 *Xylella fastidiosa* (*Xf*) primers amplified a 733-base pair (bp) region of the *Xf* genome if present [37].

## 5. Conclusions

Environmental conditions play an important role in the thermal detection of *Xfm AR3* in SHB blueberry. For the conspicuous presentation of thermal disease symptoms in greenhouse conditions, an external source of heat that increases temperatures above ambient (either a heat lamp or direct sunlight) was necessary, regardless of whether there were visible disease symptoms. Even in direct sunlight, inoculated plants displaying symptoms of BLS did not present the thermal signatures associated with BLS until the ambient air temperatures exceeded 12 °C. Despite some of these limitations, we present evidence of detectable thermal symptoms displayed prior to the emergence of subtle (non-diagnostic) BLS disease symptoms that were never observed on control plants. Furthermore, whole plant images suggested that the patchy and abnormally hot leaves on *Xfm AR3*-infected blueberry plants may be an important characteristic under field conditions prior to the display of diagnostic BLS foliar symptoms. While these assertions require more rigorous validation for BLS, especially under field conditions, we suspect that similar conspicuous within plant differences in foliar thermal temperatures (even within a single stem) may also be present for other woody plants that are infected by *Xf,* such as grapes, coffee, citrus, pecans, plums, and olives.

## Figures and Tables

**Figure 1 plants-12-03562-f001:**
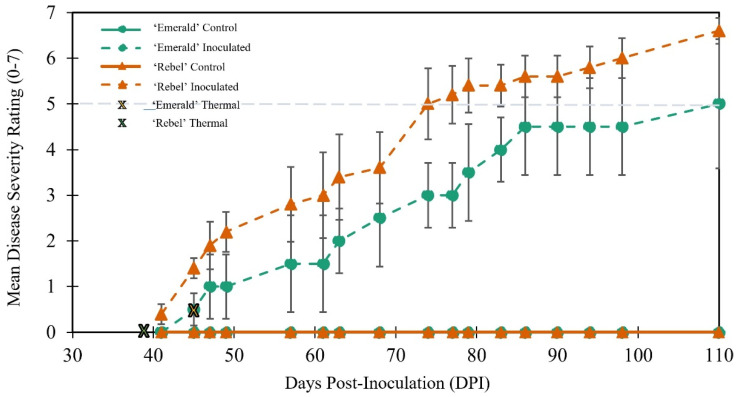
Mean disease severity (±SEM) progression for *Xylella fastidiosa* subspecies *multiplex* (*Xfm AR3*) inoculated southern highbush (SHB) blueberry over 110 days post-inoculation (DPI). The green and yellow “X”s represent the earliest thermal signatures (single leaves that were conspicuously warmer than adjacent leaves) in cultivars ‘Rebel’ and ‘Emerald’, respectively. The dotted line at mean disease severity rating “5” corresponds approximately to when *Xfm AR3* is diagnosable based on traditional visible symptoms. Note that the disease progress lines for the controls of both Emerald and Rebel cultivars overlap due to a lack of disease symptoms.

**Figure 2 plants-12-03562-f002:**
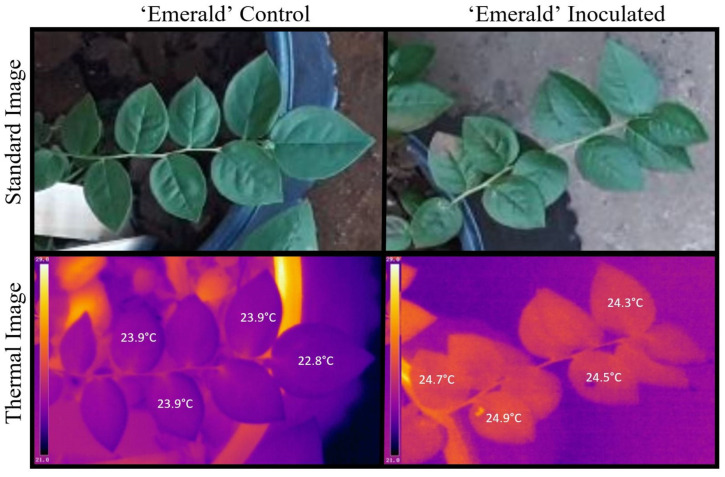
Standard and thermal images of an ‘Emerald’ buffer-only control and ‘Emerald’ *Xfm AR3* inoculated plant showing four temperature points (one on each leaf) of the inoculated stem.

**Figure 3 plants-12-03562-f003:**
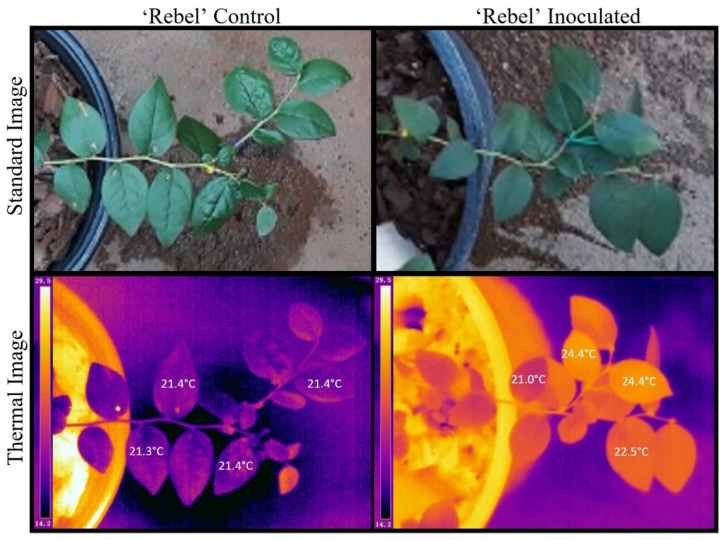
Standard and thermal images of a ‘Rebel’ buffer-only control and ‘Rebel’ *Xfm AR3* inoculated plant showing four temperature points (one on each leaf) of the inoculated stem.

**Figure 4 plants-12-03562-f004:**
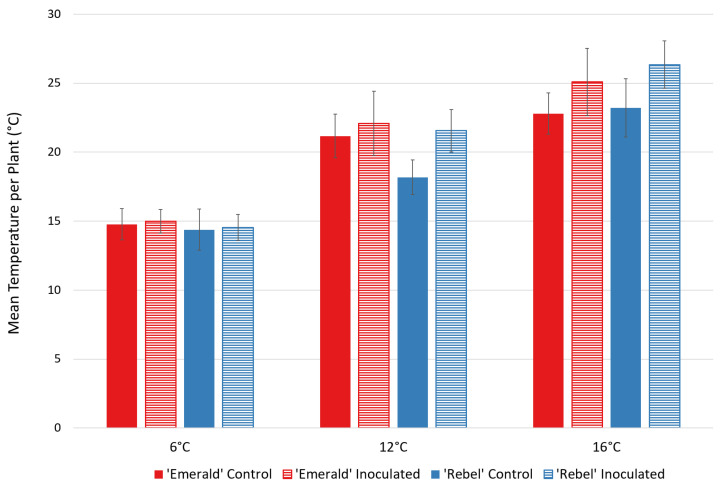
Mean temperatures (± SD) of buffer-only control and *Xfm AR3* inoculated plants for ‘Emerald’ (*n* = one control and two inoculated plants) and ‘Rebel’ (*n* = three inoculated and three control plants) blueberry cultivars at three different ambient temperatures under direct, full sun. No statistically significant mean leaf temperature differences were observed between buffer-only control and *Xfm AR3* inoculated plants at 6 °C ambient temperature for either cultivar (‘Rebel’: *t* = −0.44, *p* = 0.66, ‘Emerald’: *t* = −0.69208, *p* = 0.49). A statistically significant difference in mean foliar temperatures was observed at 12 °C ambient temperature (*t* = −3.41, *p* = 0.0015) for ‘Rebel’ (*t* = −3.41, *p* = 0.0015) but not for ‘Emerald’ (*t* = −1.42, *p* = 0.16). At 16 °C ambient temperature, both ‘Rebel’ (*t* = −5.16, *p* = 8.11 × 10^−6^) and ‘Emerald’ (*t* = −7.24, *p* = 1.14 × 10^−8^) statistically differed in mean leaf temperatures between control and inoculated plants.

**Figure 5 plants-12-03562-f005:**
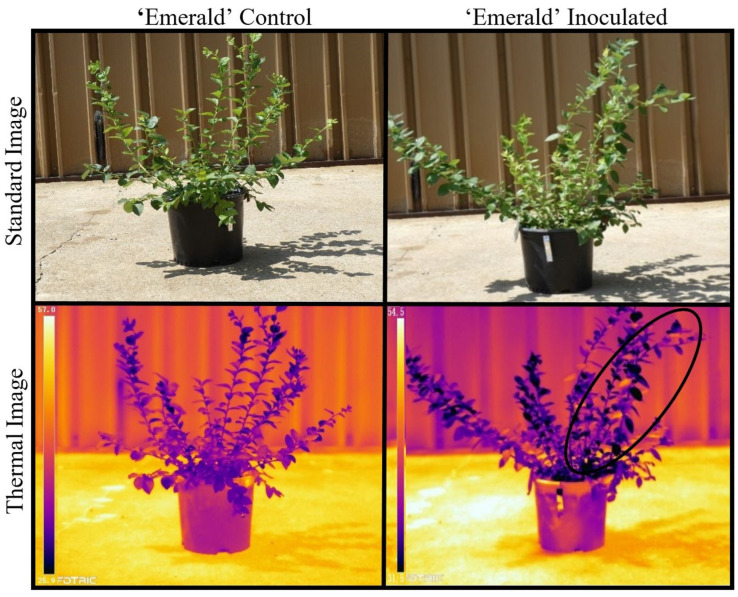
An ‘Emerald’ *Xfm AR3* inoculated plant in the standard visible spectrum (top row) and in the thermal spectrum (bottom row) displaying patchy heat accumulation when placed outside in full sun.

**Table 1 plants-12-03562-t001:** Minimum (min), maximum (max), arithmetic mean (mean ± st.dev.), median (interquartile range: 25th and 75th percentiles), and standard deviation (sd) of *Xfm AR3* inoculated ‘Emerald’ and ‘Rebel’ plants and SCP buffer-only control plants at 14, 34, and 54 days post-inoculation (DPI).

	n = No. Leaves/Plant	Min (°C)	Max (°C)	Mean (°C)	Median (°C)	sd
14 DPI						
*‘Emerald’*						
Control	10	27.0	32.7	29.1 ± 0.59	28.7 (27.9,29.4)	1.87
Inoculated	10	23.9	37.3	28.0 ± 1.23	27.0 (25.5,29.5)	3.90
*‘Rebel’*						
Control	10	24.9	29.8	27.2 ± 0.59	26.8 (25.6,28.6)	1.88
Inoculated	10	24.4	30.4	27.5 ± 0.71	27.4 (25.6,29.2)	2.25
34 DPI						
*‘Emerald’*						
Control	10	27.2	28.6	27.9 ± 0.15	27.8 (27.5,28.2)	0.47
Inoculated	10	26.3	29.9	27.5 ± 0.31	27.3 (27.0,27.6)	0.99
*‘Rebel’*						
Control	10	25.6	29.7	25.8 ± 0.75	26.0 (23.8,26.8)	2.36
Inoculated	10	28.5	32.1	30.3 ± 0.37	30.2 (29.5,31.0)	1.16
54 DPI						
*‘Emerald’*						
Control	10	27.2	31.0	28.8 ± 0.36	28.5 (28.1,29.4)	1.15
Inoculated	10	27.4	33.8	30.3 ± 0.64	30.4 (29.1,31.6)	2.03
*‘Rebel’*						
Control	10	21.6	26.7	24.5 ± 0.56	24.4 (23.5,26.1)	1.76
Inoculated	10	26.7	33.6	29.8 ± 0.69	30.2 (28.1,30.9)	2.17

**Table 2 plants-12-03562-t002:** Minimum (min), maximum (max), arithmetic mean (mean ± st.dev.), median (interquartile range: 25th and 75th percentiles), and standard deviation (sd) of *Xfm AR3* inoculated ‘Emerald’ and ‘Rebel’ plants and SCP buffer-only control plants at 151 DPI.

	n = No. Leaves/Plant	Min (°C)	Max (°C)	Mean (°C)	Median (°C)	sd
*‘Emerald’*						
Control	20	31.5	42.0	35.2 ± 0.71	34.6 (32.7,36.6)	3.16
Inoculated	20	29.9	45.2	38.7 ± 1.01	37.6 (36.2,43.9)	4.51
*‘Rebel’*						
Control	20	33.0	42.9	37.3 ± 0.55	37.1 (35.5,38.5)	2.46
Inoculated	20	35.0	46.2	39.5 ± 0.70	39.0 (36.8,41.6)	3.15

**Table 3 plants-12-03562-t003:** The disease severity rating scale was developed by Oliver et al. [36], where the proportion of the inoculated stem is categorized into a 0–7 scale.

Severity Rating	Severity Definition	Visual References
0	No visible symptoms; green leaves only	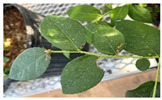
1	One or two yellowing leaves between the leaf margin and the midrib	
2	One or two yellowing leaves with reddening areas	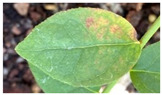
3	>2 formerly yellowing leaves have begun to turn red, but <50% of the affected stem	
4	>2 yellow leaves have turned red with marginal necrotic areas on <50% of the affected stem	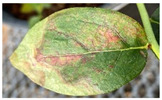
5	>50% of leaves on the affected stem are partly yellow, with red interveinal areas and necrotic areas expanding	
6	>50% of leaves on the affected stem are red or brown with increased marginal necrosis	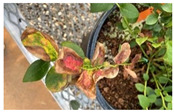
7	All leaves of the affected stem have severe leaf scorching and necrosis	

## Data Availability

Data will be made available upon request to the corresponding author.

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
