# Peer review of "Evidence of Xylella fastidiosa Infection and Associated Thermal Signatures in Southern Highbush Blueberry (Vaccinium corymbosum Interspecific Hybrids)"

_plants, 2023, doi:10.3390/plants12203562_

Round 1

Reviewer 1 Report

The manuscript (Evidence of Xylella fastidiosa infection and Associated Thermal Signatures in Southern Highbush Blueberry (Vaccinium corymbosum interspecific hybrids) is interesting but it should be improved in some points.

I do think that you must rearrange your introduction. For example, you explained that the disease is not diagnosable sometimes in the third paragraph and again in the fourth paragraph. Similarly, the statement that Xf management is difficult should not be repeated in different paragraphs. Please unify and organize the introduction.

The aims of the study should be clearly written. Please avoid to write what you did and what you expect. It seems like a proposed explanation.

Figure 1 is not easy to follow the thermal lines

Please add the figure of Polymerase chain reaction (PCR) results.

The discussion after the sentence (these observations were consistent with previous BLS studies), I strongly suggest that you should explain the results of these BSL studies compared with your results.

Please explain the statistical analysis in the end of the Materials and methods

Moderate editing is suggested specially for the introduction.

Author Response

Reviewer #1

The manuscript (Evidence of Xylella fastidiosa infection and Associated Thermal Signatures in Southern Highbush Blueberry (Vaccinium corymbosum interspecific hybrids) is interesting but it should be improved in some points.

I do think that you must rearrange your introduction. For example, you explained that the disease is not diagnosable sometimes in the third paragraph and again in the fourth paragraph. Similarly, the statement that Xf management is difficult should not be repeated in different paragraphs. Please unify and organize the introduction.

Thank you for the comments.  We were attempting to write for a broad audience that represents the Plants readership, as opposed to strictly Plant Pathologists, so the 3rd paragraph mentioned by the reviewer was our attempt to demonstrate that there is a general diagnosis pattern and management philosophy that underpins systemic plant diseases. We do not introduce BLS in this third paragraph intentionally. In the 4th paragraph we introduce BLS and explain how it shares the same problems with systemic plant diseases (which were highlighted in the previous paragraph). To make this distinction between a typical plant systemic disease (3rd paragraph) and BLS specifically (4th paragraph), we added the words “In general” to several key sentences in the 3rd paragraph which should clearly indicate to the reader that we are providing information about systemic plant diseases in a broad sense. Hopefully, these changes will make it clear to the reader that BLS is similar to other systemic plant diseases in important ways. We felt it was important to write the manuscript with this approach because it suggests that our problems with BLS are not specifically BLS only issues and that our methods may be useful for other Xf caused diseases. To reasonably make this claim we draw parallels between BLS and other systemic plant diseases.

The aims of the study should be clearly written. Please avoid to write what you did and what you expect. It seems like a proposed explanation.

We are somewhat confused about this comment as the Introduction did attempt to explain our biological rationale for investigating the role of thermal imaging in early disease detection from what is known about Xf caused plant diseases. In the last sentence of the Introduction we have a hypothesis statement which reads, “If Xylella fastidiosa infection interferes with evapotranspirative leaf cooling, we expect to observe hotter leaves on experimentally inoculated blueberry plants when compared to control plants, either before or coinciding with subtle foliar disease symptom appearance (slight leaf discoloration).”                                                                                                                                                                            

Figure 1 is not easy to follow the thermal lines

We understand that the control lines overlap each other because we did not record any disease symptoms. It was difficult to select a visual approach that was consistent between all of the disease development curves without appearing to look odd.  This is the figure we settled on. We have made an addition to the Figure 1 caption which reads, “Note that the disease progress lines for the controls of both Emerald and Rebel cultivars overlap due to a lack of disease symptoms.”

Please add the figure of Polymerase chain reaction (PCR) results.

Unfortunately, we no longer have this image file so we could not include it in the revision.

The discussion after the sentence (these observations were consistent with previous BLS studies), I strongly suggest that you should explain the results of these BSL studies compared with your results.

We attempted to do exactly this by acknowledging in each paragraph the specific weaknesses in our study and how the results we observed were consistent with what has been recorded with BLS and other Xf caused diseases. We know that our study has weaknesses which prevent a more confident and conclusive statement, so we pointed out how our results were consistent with past research. As coauthors we wished to fairly present our results but not over interpret those results and allow the reader to weigh the drawbacks versus our data and the published literature. We also tried to clarify that the visible disease symptom progression was consistent with previous studies (as opposed to thermal symptoms – which this is the first study to detail these apparent symptoms).

Please explain the statistical analysis in the end of the Materials and methods

We opted to explain the statistical analysis at the end of each subsection in the methods which addressed separate experiments because this was more efficient and straightforward than redescribing each experiment and analysis again in a separate analysis section.  We added the statistical program package used to run the analyses in the revision.

Comments on the Quality of English Language

Moderate editing is suggested specially for the introduction.

Reviewer 2 Report

Guzmán Martinez et al present a novel diagnostic tool for early detection of BLS of blueberry, basen on the concept  that early detection and treatment appear to be key in the control of disease outbreaks. The manuscript is very well written and structured. 
Authors take advantage that before BLS leaf scorching symptoms appear, the plant may experience local water translocation dysfunction and use thermal imagen as diagnostic tool.

The methods applied support enough the finding, in my opinion this manuscript is of high interest to the readers of plants and should be published after minimal revision.

The manuscript is easy to read and don’t have serious issues of language, 

Author Response

Reviewer #2

Guzmán Martinez et al present a novel diagnostic tool for early detection of BLS of blueberry, basen on the concept  that early detection and treatment appear to be key in the control of disease outbreaks. The manuscript is very well written and structured. 

Authors take advantage that before BLS leaf scorching symptoms appear, the plant may experience local water translocation dysfunction and use thermal imagen as diagnostic tool.

The methods applied support enough the finding, in my opinion this manuscript is of high interest to the readers of plants and should be published after minimal revision.

Thank you.  We have also gone through the entire manuscript and made small editorial changes to fix typos, some style conflicts, and any grammatical issues that we did not catch in the submitted version. The more substantial changes were highlighted in yellow.

Round 2

Reviewer 1 Report

I don't agree with the authors and I think the introduction should be rearranged as I mentioned before. For example, the author could transfer the paragraph (BLS is not accurately diagnosable through visible symptoms until disease progression has advanced to the point of stem discoloration and significant levels of leaf discoloration and leaf drop. By the time BLS is visibly diagnosable, Xf is likely to have spread from the source plant to others through insect carriers) to the following paragraph which is explaining disease symptoms also.

Please clearly explain your aims of study.

Additionally, it is better for clarity to add a paragraph about of statistical analysis.

The English is fine.

Author Response

The original comment:

"I don't agree with the authors and I think the introduction should be rearranged as I mentioned before. For example, the author could transfer the paragraph (BLS is not accurately diagnosable through visible symptoms until disease progression has advanced to the point of stem discoloration and significant levels of leaf discoloration and leaf drop. By the time BLS is visibly diagnosable, Xf is likely to have spread from the source plant to others through insect carriers) to the following paragraph which is explaining disease symptoms also."

Thank you for the suggestion of merging the topics among the two paragraphs in the Introduction into one paragraph.

We have done as suggested and it did considerably shorten the introduction and make the points more concise.  We hope that this change was how the reviewer intended the Introduction to be modified.  Because we omitted the majority of one paragraph and a portion of the other paragraph, we do not show the omissions but do indicate the revised paragraph that is highlighted in yellow.

The new paragraph now reads as follows:

"Xylella fastidiosa (Xf), a gram-negative bacterium, is an important and potentially devastating plant pathogen. Like other plant pathogens that cause systemic disease, the earliest visible symptoms of Xf infections can be very subtle (slight leaf discoloration, < 1 cm diameter brown regions, rolled and browning leaf margins). These same symptoms may be readily confused with abiotic stresses such as water deficit, fertilizer and/or chemical burns, insect herbivory, and mechanical damage [18]. Xf caused diseases are managed like many other systemic plant diseases. The disease management strategy is to remove the obviously infected plants and to consider removing plants within a buffer zone which could be latently infected [19,20]. Because Xf caused diseases are typically slow to develop (at least one growing season), vectoring of Xf to uninfected plants via feeding by flying, hemipteran insects (primarily Cicadellidae and Aphrophoridae [21–25]), almost assuredly spreads Xf to uninfected hosts far beyond the immediate disease outbreak area. Unfortunately, the spatial patterns of disease transmission are difficult to understand, and therefore manage, due to a combination of poorly understood insect vector behaviors and a long disease latent period. Xf causes disease on hundreds of different plant species throughout the warmer, temperate regions of the world [26], including culturally important plants such as grapes (Pierce’s disease), olives (olive quick decline syndrome), and coffee (coffee leaf scorch) [27,28]. So, there is a general interest in methods to detect disease as early as possible, which could improve plant removal strategies to more effectively limit disease spread. In the southeastern United States, bacterial leaf scorch (BLS) of blueberry is caused by Xf subsp. multiplex and Xf subsp. fastidiosa [29,30]. BLS causes significant yield loss and plant death in this region which produces a large proportion of the spring North American fresh market blueberries. There are no curative treatments for BLS (e.g.  antibiotics) and BLS appears typical among Xf caused diseases in having a long, potentially multiyear latent period [31,32, 33]."